# Incomptine A Induces Apoptosis, ROS Production and a Differential Protein Expression on Non-Hodgkin’s Lymphoma Cells [note 1]

**DOI:** 10.3390/ijms221910516

**Published:** 2021-09-29

**Authors:** Emmanuel Pina-Jiménez, Fernando Calzada, Elihú Bautista, Rosa María Ordoñez-Razo, Claudia Velázquez, Elizabeth Barbosa, Normand García-Hernández

**Affiliations:** 1Posgrado en Ciencias Biológicas, Universidad Nacional Autónoma de México, Ciudad Universitaria 3000, Coyoacán, Mexico City CP 04510, Mexico; epinaj@hotmail.com; 2Unidad de Investigación Médica en Genética Humana, UMAE Hospital Pediatría 2° Piso, Centro Médico Nacional Siglo XXI, Instituto Mexicano del Seguro Social, Av. Cuauhtémoc 330, Col. Doctores, Mexico City CP 06725, Mexico; romaorr@yahoo.com.mx; 3Unidad de Investigación Médica en Farmacología, UMAE Hospital de Especialidades, 2° Piso CORSE, Centro Médico Nacional Siglo XXI, Instituto Mexicano del Seguro Social, Av. Cuauhtémoc 330, Col. Doctores, Mexico City CP 06725, Mexico; 4CONACYT—Consorcio de Investigación, Innovación y Desarrollo para las Zonas Áridas, Instituto Potosino de Investigación Científica y Tecnológica A.C., San Luis Potosí CP 78216, Mexico; francisco.bautista@ipicyt.edu.mx; 5Área Académica de Farmacia, Instituto de Ciencias de la Salud, Universidad Autónoma del Estado de Hidalgo, Km 4.5, Carretera Pachuca-Tulancingo, Unidad Universitaria, Pachuca CP 42076, Mexico; cvg09@yahoo.com; 6Sección de Estudios de Posgrado e Investigación, Escuela Superior de Medicina, Instituto Politécnico Nacional, Salvador Díaz Mirón esq. Plan de San Luis S/N, Miguel Hidalgo, Casco de Santo Tomas, Mexico City CP 11340, Mexico; rebc78@yahoo.com.mx

**Keywords:** incomptine A, sesquiterpene lactone, *Decachaeta incompta*, cytotoxic activity, iTRAQ, apoptosis, ROS production

## Abstract

Sesquiterpene lactones are of pharmaceutical interest due their cytotoxic and antitumor properties, which are commonly found within plants of several genera from the Asteraceae family such as the *Decachaeta* genus. From *Decachaeta incompta* four heliangolide, namely incomptines A-D have been isolated. In this study, cytotoxic properties of incomptine A (**IA**) were evaluated on four lymphoma cancer cell lines: U-937, Farage, SU-DHL-2, and REC-1. The type of cell death induced by **IA** and its effects on U-937 cells were analyzed based on its capability to induce apoptosis and produce reactive oxygen species (ROS) through flow cytometry with 4′,6-diamidino-2-phenylindole staining, dual annexin V/DAPI staining, and dichlorofluorescein 2′,7′-diacetate, respectively. A differential protein expression analysis study was carried out by isobaric tags for relative and absolute quantitation (iTRAQ) through UPLC-MS/MS. Results reveal that **IA** exhibited cytotoxic activity against the cell line U-937 (CC_50_ of 0.12 ± 0.02 μM) and the incubation of these cells in presence of **IA** significantly increased apoptotic population and intracellular ROS levels. In the proteomic approach 1548 proteins were differentially expressed, out of which 587 exhibited a fold-change ≥ 1.5 and 961 a fold-change ≤ 0.67. Most of these differentially regulated proteins are involved in apoptosis, oxidative stress, glycolytic metabolism, or cytoskeleton structuration.

## 1. Introduction

The non-Hodgkin’s lymphomas (NHL) are a heterogeneous group of illnesses that arise primarily in the lymph nodes or other lymph tissues due to malignant transformation of B, T, and NK lymphocytes. In 2015, nearly 4.3 million people around the world were reported with NHL, which accounted for 231,400 deaths. In Mexico until 2017, NHL were the fourth cause of mortality between patients with cancer [1]. The anticancer drugs commonly used alone or combined for the treatment of NHL include cyclophosphamide, prednisone, cisplatin, methotrexate, and doxorubicin as well as vincristine and etoposide. All these drugs show strong side effects. In Mexico, methotrexate is regarded as the preferred drug, however, in most cases, high dose methotrexate administration required for lymphoma treatment leads to genotoxic damage and the appearance of side effects including acute kidney injury, nephrotoxicity, myelosuppression, mucositis, and hepatotoxicity as well as dermatologic toxicity [1,2]. Therefore, there is a need to search and develop new, safer, and more effective anticancer agents.

Incomptine A (**IA**) is a heliangolide-type sesquiterpene lactone isolated from leaves of *Decachaeta incompta* [3]. The study of sesquiterpene lactones (**SL**) has led to discover the molecular mechanisms involved in the cytotoxicity of these compounds which include reduced glutathione depletion, prevented NF-κB activation, increased intra-cellular reactive oxygen species levels, and downregulation of Bcl-2 anti-apoptotic proteins, as well as intrinsic apoptosis activation [4]. Furthermore, the SL can inhibit glycolytic enzymes [5], leading to energy metabolism disruption and compromising cell viability [6]. **IA** has a 3-methylenedihydrofuran-2(3H)-one moiety, to which the cytotoxic properties of sesquiterpene lactones have been associated, due to its capability to act as a Michael acceptor and react with sulfhydryl residues of proteins. Previously, it was reported that **IA (**Figure 1) induces energy metabolism disruption as result of downregulation of the glycolytic enzymes enolase and fructose biphosphate aldolase in *Entamoeba histolytica* trophozoites [7], and exhibit a broad range of biological properties including antiprotozoal, antibacterial, trypanocidal, phytotoxic, and spermatic activities as well as anti-propulsive properties [3,8,9]. Herein, we examined the cytotoxic effects of **IA** against U-937 cells including apoptosis induction, and reactive oxygen species (ROS) production, as well as the differential protein expression.

## 2. Results and Discussion

Non-Hodgkin’s lymphoma represents a wide spectrum of illnesses that are a significant cause of morbidity and mortality around the world. Chemotherapy is an effective treatment against NHL that is used alone or combined including methotrexate, cyclophosphamide and doxorubicin. However, until today all the drugs used induced various side effects such is the case of methotrexate that is the drug of choice in Mexico [1,2,7]. In an effort to improve the therapy of cancer, specifically NHL, the study of specialized metabolites isolated from medicinal plants such as sesquiterpene lactones [4] constitute a source of potential new anti-lymphoma compounds with low toxicity. Based on these facts, the above prompted to assay the cytotoxic effects of **IA** and explore its mechanism of action.

In the present work, the cytotoxic activities of incomptine A (**IA**) against four subtypes of NHL cell lines, U-937 (diffuse histiocytic lymphoma), Farage (diffuse large B-cell non-Hodgkin’s lymphoma), SU-DHL-2 (diffuse large B-cell lymphoma), and REC-1 (mantle cell lymphoma), were evaluated by DAPI staining for 24 h using methotrexate as a drug control and flow cytometry analysis. The results (Table 1) revealed that all subtypes of NHL human cells used were susceptible to incomptine A (**IA**) with CC_50_ values ranging from 0.12 ± 0.02 to 3.5 ± 0.01 μM, being most cytotoxic against the U-937 cell line. In this case, incomptine A (**1**) was four-fold more active than methotrexate, used as drug control. Additionally, the use of high doses of methotrexate has been reported to be toxic to humans [2] and mice [10]. These results suggest that **IA** has antitumoral potential [4,11,12,13].

In addition, because DAPI staining was used, the strong cytotoxic properties of **IA** can be associated with an apoptotic process [11,12]. To explore the possible mechanism of action of cell death induced on U-937 cell lines. Cells were incubated with a concentration equivalent to CC_50_ value of **IA** (0.12 μM) or MTX (0.5 μM) for 24 h, and were then harvested, stained with annexin V/DAPI and further analyzed by flow cytometry. The distribution of cells resulting from this experiment is shown in Figure 2A. U-937 cells treated with **IA** or MTX showed a decrease in the percentage of viable cells with 39.9% and 49.4%, respectively, compared to control (84.3%). Additionally, the results showed an important early apoptotic effect on U-937 cells caused by incomptine A (**IA**), it was closer than MTX (Figure 2B).

The production of reactive oxygen species (ROS) has been associated with apoptosis induction under physiologic and pathological conditions [13,14,15]. Previous studies carried out with other SL indicate that cytotoxic, antitumor, and anticancer activities are associated to the presence of a 3-methylenedihydrofuran-2 (*3H*)-one moiety [4]. Therefore, interaction of this moiety with reduced glutathione (GSH), leads to its depletion and in consequence, the accumulation of reactive oxygen species that damage cell biomolecules such as lipids, structural proteins and DNA, as well as cause the initiation of mitochondria-dependent apoptosis pathway [4,16]. Based on the above, the production of ROS induced by **IA** was also measured. The results showed (Figure 3A) that DCF oxidation shifts the fluorescence peak towards the right, which is proportional to ROS production. Compared to positive control (H_2_O_2_), compound **IA** slightly shifts the DCF fluorescence peak, and MTX was inactive. Mean fluorescence intensity of DCF was recorded for each treatment and subsequently the DCF index was calculated with respect to control (Figure 3B). We observed that compound **I**A and H_2_O_2_ had significant differences in DCF oxidation, whereas MTX showed no changes. The cytotoxic activity, apoptosis induction, and increase of reactive oxygen species suggest that **IA** induces oxidative stress that finally leads to apoptosis on U-937 cells [4,16], unlike methotrexate where endogenous ROS levels did not increase, suggesting that incomptine A (**IA**) and MTX have a different action mechanism.

To understand the cytotoxic mechanism of **IA** on U-937 cells, a proteomic approach was carried out, in which we analyzed the differential protein expression by isobaric tags for relative and absolute quantitation (iTRAQ) coupled with liquid chromatography–tandem mass spectrometry (LC-MS/MS) [17]. U-937 cells were treated with incomptine A (0.12 μM) or methotrexate (0.5 μM) for 24 h and non-treated cells were used as control. We identified and quantified 3222 proteins for all evaluated conditions. Normalized reporter values were used to calculate the ratios of each treatment with respect to control for all proteins [17,18]. Fold change criteria for upregulated proteins were established for those with ratios ≥ 1.5 and downregulated for proteins with ratios ≤ 0.67. Differentially expressed proteins were analyzed using STRING Functional Enrichment Analysis to identify biological processes and pathways that were altered by administration of incomptine A (**IA**) [17,18,19]. Enrichment analysis revealed 82 significantly enriched biological processes and signaling pathways (*p* < 0.05), but only those with percentages of proteins involved in each process greater than 5% were plotted in Figure 4. We observed that **IA** modifies the expression of proteins involved in apoptosis, oxidative stress response, DNA repair, cell cycle checkpoints, glycolysis, cytoskeleton organization, NF-κB signaling and autophagy (Table 2); whereas, the MTX alter the expression of proteins implicated in purine nucleotide biosynthesis related pathways, G2/M checkpoint and intrinsic apoptosis. Proteomic analysis was validated by identifying the reported antitumor mechanism of methotrexate, which depends on dihydrofolate reductase inhibition to prevent catalysis of dihydrofolate to tetrahydrofolate, an essential cofactor in purine and thymidylate biosynthesis, which are crucial events for cell division and proliferation [20]. Exhausted nucleotide biosynthesis is associated with a poor prognosis of neoplasms of breast cancer by action of guanosine-5-triphosphate (GTP), that generates guanosine monophosphate (GMP), which is required for cell proliferation by MAPK pathway [21]. Our data confirmed that biological processes such as GMP, nucleobase, nucleoside and purine ribonucleoside monophosphate biosynthesis were differentially expressed by MTX treatment on U-937 cells, intrinsic apoptosis signaling, DNA synthesis and G2/M checkpoint [22]. The activity of MTX seems to vary among different cell groups, since in lymphoid cell lines it increases reactive oxygen species, leading to DNA damage, activation of cell cycle arrest and DNA repair mechanisms whereas oxidative stress induction occurs to a lesser extent in monocytic cells [23,24], which was confirmed in our studied U-937 histiocytic lymphoma monocytes.

The treatment with incomptine A (**IA**) differentially expressed changes in the amounts of proteins involved in process cells such as apoptotic mitochondrial fragmentation, apoptosis, detoxification of ROS, cell cycle checkpoints, DNA repair, autophagy, canonical glycolysis, pyruvate metabolism and cytoskeleton organization. Upregulation of pro-apoptotic proteins BAX, PDCD4 and NDRG1 in our U-937 cells indicates modification of mitochondrial membrane associated with activation of intrinsic apoptosis [25]. Glutathione transferases are involved in chemoresistance and have been studied as molecular targets. These enzymes can reduce glutathione levels or inhibit their function to prevent detoxification generated by chemotherapy [26], leading to generation of oxidative stress which induces DNA damage and apoptosis [27].

We identified downregulation of glutathione transferases caused by incomptine A (**IA**), and some of them were microsomal glutathione *S*-transferase and glutathione *S*-transferase kappa 1, upregulation of peroxiredoxin oxidative stress response transferases PRDX1, PRDX2 and PRDX6. Oxidative stress induced by **IA**, also led to upregulation of DNA repair protein MSH2, involved in 8-oxoguanine repair [27,28], and autophagic proteins ATG3, ATG7, LAMP1 and LAMP2 on U-937 cells, coinciding with previously reported autophagic activity of sesquiterpene lactones [29].

According to the results obtained, the oxidative stress observed in U-937 cells by ROS production, and the subsequent proteomic analysis suggested that a mechanism involving NF-κB is not possible due to NF-κB signaling p105 and p65 were found upregulated. NF-κB is known for being upregulated in cancer, conferring tumor cells the ability to evade apoptosis through the expression of Bcl-2 family anti-apoptotic proteins [30,31]. Anti-apoptotic Bcl-2 family members were not detected dysregulated, only the Bcl-2 associated transcription factor 1 was found downregulated, which confirms that **IA** does not interact with NF-κB on U-937 cells.

On the other hand, aerobic glycolysis is known to be abnormally activated in non-Hodgkin’s lymphoma, conferring drug resistance to tumor cells, consequently glycolytic enzymes are also considered as drug development targets [32]. Interestingly, the proteomic analysis revealed differential expression of glycolysis pathway, being lactate dehydrogenase A (LDHA), lactate dehydrogenase B (LDHB) and fructose-biphosphate aldolase A (ALDOA), the identified downregulated glycolytic enzymes by (**IA**) treatment. ALDOA is a key glycolytic enzyme, the high expression of which has been associated to tumor progression and poor prognosis in hepatocellular carcinoma [33]. Glycolytic enzymes play a physiological role that is not limited to catalytic function and can participate as regulators of cellular processes, such as actin filaments organization, p53 signaling, and cell cycle progression, considered as necessary cellular events for survival and proliferation of cancer cells. When ALDOA-actin cytoskeleton interaction is perturbed, significant elevation of ROS production leads to decreased ATP synthesis, increased calcium levels and activates caspases [34]. Since **IA** also downregulated cytoskeleton elements such as alpha actinin-4 and tropomyosin alpha-4 chain, additional studies must be realized to know how compound **IA** treatment triggers the activation of this mechanism related to regulatory function of ALDOA, and how the downregulation of ALDOA and its glycolytic activity is associated to the induction of oxidative stress. To our knowledge, this is the first report that the inhibition of glycolytic enzymes (ALDOA, LDHA, and LDHB) and cytoskeleton proteins (alpha actinin-4 and tropomyosin alpha-4 chain) may be part of the mechanism by which SL produce their antitumor effects.

In order to visualize the different clusters involved in pro-apoptotic activity of **IA**, we performed a Protein–Protein Interaction Analysis (PPI) of differentially expressed proteins (Figure 5) and the results obtained suggest that the possible action mechanism may be the apoptosis activation, where glycolytic enzymes ALDOA, LDHA and LDHB were biologically connected to pro-apoptotic proteins as BAX and BID, as well to the glutathione transferases MGST1 and GSTK1 [33,34].

## 3. Conclusions

This study used cytotoxic effects of incomptine A (**IA**) against U-937 cells including apoptosis induction, and reactive oxygen species (ROS) production, followed by iTRAQ labeling approach to determinate its potential as an antitumor agent. Our findings suggest that **IA** is a sesquiterpene lactone with antitumor potential that induces apoptosis by redox imbalance [4] on U-937 cells. In addition, antitumor potential could be associated with inhibition of glycolytic enzymes such as LDHA, LDHB and ALDOA. Future systematic studies will investigate how **IA** regulates the expression of glycolytic enzymes and its correlation with apoptosis induction and ROS production. Research should be directed to confirm the antitumor potential of **IA** in non-Hodgkin’s lymphoma using animal models [1].

## 4. Materials and Methods

### 4.1. Incomptine A (IA) Isolation

Compound **IA** was isolated from the aerial parts of *Decachaeta incompta* (syn.: *Eupatorium incomptum*; Asteraceae) collected in the State of Oaxaca, Mexico. The plant was identified by the MS Abigail Aguilar Contreras taxonomist of the Instituto Mexicano del Seguro Social (IMSS). A voucher specimen (15311) was deposited in the Herbarium IMSSM. The extraction and isolation procedure were performed according the protocol previously described [3]. Identification of **IA** was made by comparison (NMR, TLC, and HPLC-DAD) with an authentic sample, having a purity near to 99%. Dimethyl sulfoxide (DMSO) was used to dissolve **IA** and methotrexate (PISA pharmaceutical).

### 4.2. Chemicals

TMT10plex Isobaric Label Reagent Set, Pierce Quantitative Colorimetric Peptide Assay (Thermo Fisher Science, Waltham, MA, USA). Triethylammonium bicarbonate buffer (1.0 M, pH 8.5 ± 0.1), Tris (2-carboxyethyl) phosphine hydrochloride solution (0.5 M, pH 7.0), iodoacetamide (IAA), formic acid (FA), acetonitrile (MeCN), methanol (MeOH), and trypsin from bovine pancreas (Promega, Madison, WI, USA) were used. Ultrapure water was obtained from a Millipore purification system.

### 4.3. Cell Cultures and Reagents

NHL cell lines: U-937 (histiocytic lymphoma, cat. CRL-1593.2); SU- DHL-2 (large B-cell lymphoma, cat. CRL-2956); Farage (B-cell lymphoma, cat. CRL-2630), and REC-1 (mantle B-cell lymphoma, cat. CRL-3004) were obtained from the American Type Culture Collection (ATCC). Cells were cultured in RPMI 1640 culture medium (GIBCO Cat: 11875–093), added with 5% fetal bovine serum (GIBCO Cat: 16000044) in a 37 °C incubator and 5% CO_2_. Cell cultures were maintained at a density of 2 × 10^6^ cells in T75 flasks.

#### 4.3.1. Cytotoxic Activity

Concentration that kills 50% cell population (CC_50_) was determined in NHL cell lines after 24 h upon exposure to **IA** by flow cytometry viability analysis. In total, 50,000 cells were seeded in 500 μL of RPMI 1640 medium supplemented with fetal bovine serum in 48-well culture plates. Cells were allowed to grow for 4 h prior to exposure to different concentrations (0.01, 0.1, 1, and 10 μM) of **IA** or methotrexate (0.01, 0.1, 1, and 10 μM). DMSO treatments were used as solvent control to rule out that cell death is due to its effect. A working solution of **IA** was prepared with RPMI 1640 and DMSO. A total of 10,000 events per sample were registered after 24 h of treatment on a FACS Verse cytometer (Beckton Dickinson) equipped with filters 404, 488, and 561 nm. DAPI was excited by the 404 nm laser and emission light was collected with filter 450/50 BP [35,36]. Percentage of mortality (DAPI positive cells) was recorded for each treatment and linearized on Excel software to obtain CC_50_.

#### 4.3.2. Annexin V/DAPI Apoptosis Assay

Apoptotic cell death index induced on U-937 cell line by **IA** was determined by flow cytometry using the Annexin V/DAPI apoptosis detection kit (BD Bio-sciences), according to the manufacturer instructions. In total, 50,000 cells were seeded in 500 μL of RPMI 1640 medium supplemented with fetal bovine serum in 48-well culture plates. Cells were allowed to grow for 4 h prior to exposure to the concentrations of **IA** (0.12 μM) or methotrexate (0.5 μM). Methotrexate was used as a drug control. DMSO was used as solvent control to rule out that cell death. A total of 10,000 events of each treatment were registered after 24 h of treatment administration on a FACS Verse cytometer (Beckton Dickinson). Relative percentages of apoptotic cells were recorded for each treatment [35,36].

#### 4.3.3. Reactive Oxygen Species Assay

Oxidative stress induced on U-937 cell line by **IA** was evaluated by flow cytometry. The increase of reactive oxygen species levels was detected based on the 2′7′dichlorofluorescin di-acetate (DCF) oxidation to 2′7′dichlorofluorescin assay. In total, 50,000 cells were seeded in 500 μL of RPMI 1640 medium supplemented with fetal bovine serum in 48-well culture plates and treated for 30 min with **IA** (0.12 μM) or methotrexate (0.5 μM). DMSO treatment was used as solvent control to rule out that ROS increased levels is due to its effect at equivalent concentration of incomptine A (**IA**) CC_50_. A total of 50 μM hydrogen peroxide (H_2_O_2_) was used as a positive control of increasing ROS levels agent and a negative control without treatment were used. DCF mean fluorescence intensity (MFI) of DCF were recorded to perform comparisons between treatments [36].

#### 4.3.4. Cell Lysis and iTRAQ-Based Proteomic Analysis

First, 1 × 10^8^ U-937 cells were harvested in 50 mL of RPMI 1640 medium added with 5% fetal bovine serum (GIBCO Cat: 16000044) at 37 °C and 5% CO_2_. Amounts necessary to obtain final concentration of **IA** (0.12 μM) or methotrexate (0.5 μM) were added and the resulting mixtures were incubated for 24 h. Non-treated cells were used as negative control. After the treatment, cells were centrifugated at 2500 rpm × 3 min and resulting pellets were washed with 1 mL of PBS and lysed with 300 μL of lysis buffer (8 M urea, 0.8 M NH_4_HCO_3_, pH 8.0) and TissueLyser. After cell lysis the tubes were centrifuged at 12,000 rpm for 15 min at 4 °C. The supernatants were collected, and protein concentration was measured using the Pierce BCA protein assay kit. Then, 100 μg of proteins per sample were reduced with 5 μL of 10 mM TCEP for 1 h at 56 °C, and subsequently alkylated with 20 μL of 20 mM IAA for 1 h at room temperature in the dark. Then, free trypsin was added into the protein solution at a ratio of 1:50. The resulting solution was incubated at 37 °C overnight and then lyophilized. The residues were re-dissolved with TEAB (100 mM). After, 20 μL of anhydrous MeCN was added to each tube and shaken for 5 min at room temperature. Each sample was centrifugated to obtain a supernatant, then the samples were transferred to the iTRAQ (TMT10 plex reagent, Thermo Fisher Science) reagent vial as follows: C, TMT10-127C; MTX, TMT10-128C, and IA, TMT10-130N. The reaction mixtures were incubated 1 h at room temperature. The reaction was quenched by addition of 8 mL of hydroxylamine 5% and incubated again for 15 min and transferred to a new microcentrifuge tube

#### 4.3.5. Nano LC-MS/MS Analyses

The iTRAQ labeled peptides were analyzed by LC-MS/MS on an Ultimate 3000 nano UHPLC system (Thermo Fisher Scientific) coupled to a Q Exactive HF mass spectrometer (Thermo Scientific) equipped with a nano spray flex ion source (Thermo Scientific). Briefly, 2 μg dissolved of each sample were injected onto a trap (PepMap C18, 100 Å, 100 μm × 2 cm, 5 μm) column, followed by fractionation on an analytical (PepMap C18, 100 Å, 75 μm × 50 cm, 2 μm) column. Linear gradients of 5–7% solvent B in 2 min, from 7% to 20% solvent B in 80 min, from 20% to 40% solvent B in 35 min, then from 40% to 90% solvent B in 4 min at 200 nL/min flow rate. Solvent A consisted of 0.1% formic acid in water and solvent B contained 0.1% formic acid in 80% Me CN. For TMT-labeled samples, the full scan was performed between the window *m*/*z* 350–1650 *m*/*z* at the resolution 120,000 at 200 Th. The automatic gain control target for the full scan was set to 3 to 6. The MS/MS scan was operated in Top 15 mode using the following settings: resolution 30,000 at 200 Th; automatic gain control target 1e5; normalized collision energy at 32%; isolation window of 1.2 Th; charge sate exclusion: unassigned, 1, >6; dynamic exclusion 40 s.

#### 4.3.6. Protein Identification

The raw MS files of six replicates were analyzed and compared against the human protein database using Maxquant (1.6.2.6). The parameters were set as follows: the protein modifications were carbamidomethylation (C) (fixed), oxidation (M) (variable), TMT-10Plex; the enzyme specificity was set to trypsin; the maximum missed cleavages were set to 2; the precursor ion mass tolerance was set to 10 ppm, and MS/MS tolerance was 0.6 Da.

#### 4.3.7. Differential Protein Analysis

Generated data were exported to an Excel file containing the 3222 identified and quantified proteins. Normalized TMT reporter values C (TMT10-127C), MTX (TMT10-128C), and IA (TMT10-130N) were used to determine protein differential expression between treatments, establishing the following ratios: MTX/C, and IA/C. The fold change criteria for upregulated proteins were established for those with ratios greater than ≥1.5 and downregulated for those proteins with ratios lower than ≤0.67. Differentially expressed proteins for each comparison were analyzed using STRING Functional Enrichment Analysis, Protein–Protein Interaction Analysis (PPI) and Kyoto Encyclopedia of Genes and Genomes (KEGG).

#### 4.3.8. Statistical Analysis

Results are expressed as the mean values ± standard error of the mean. Statistical analyses were performed by using Excel version 16.46 for Macintosh. The statistical evaluation was carried out through an analysis of variance followed by Dunnet’s test for multiple comparisons. *p* < 0.05 (*) was considered a statistically significant difference between the group means.

## Figures and Tables

**Figure 1 ijms-22-10516-f001:**
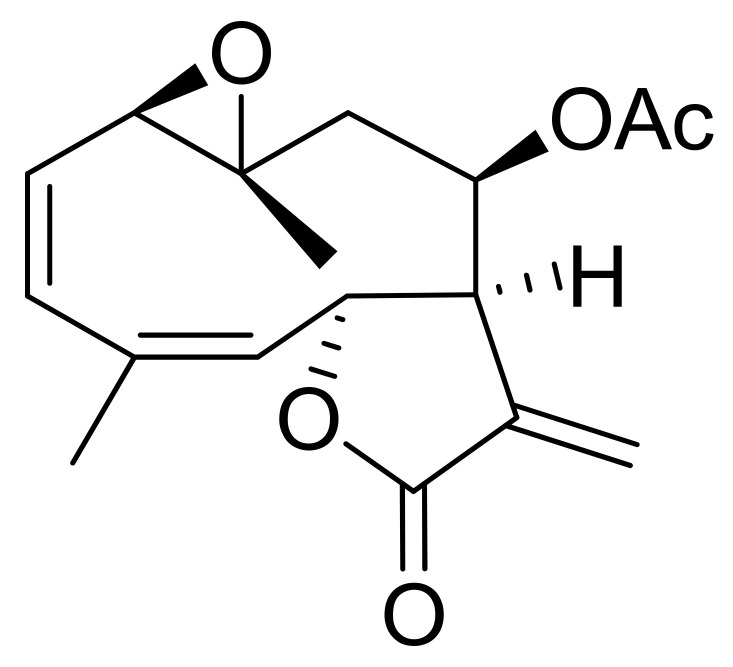
Structure of incomptine A (**IA**).

**Figure 2 ijms-22-10516-f002:**
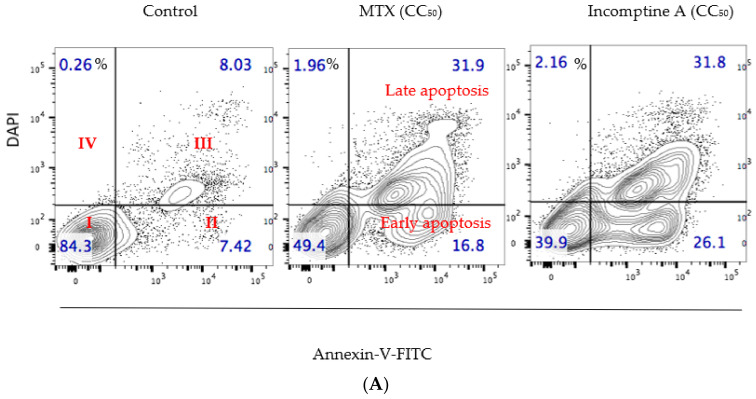
Apoptosis induced on U-937 cells by incomptine A (**IA**) and methotrexate (MTX). U-937 cells were treated with CC_50_ (mM) of compounds for 24 h. (**A**) Representative images of contour maps from evaluated treatments with relative death percentage for each quadrant (red numbers). Quadrant I annexin V−/DAPI−; quadrant II annexin V+/DAPI− early apoptosis; quadrant III annexin V+/DAPI+ late apoptosis, and quadrant IV annexin V−/DAPI−. Apoptotic population increases in IA and MTX CC_50_ treatments, revealing apoptosis induction. (**B**) Histograms of apoptosis were calculated respectively to negative control, finding significant differences in incomptine A and MTX treatments. Data is expressed as means ± SEM, *n* = 3; * *p* < 0.05.

**Figure 3 ijms-22-10516-f003:**
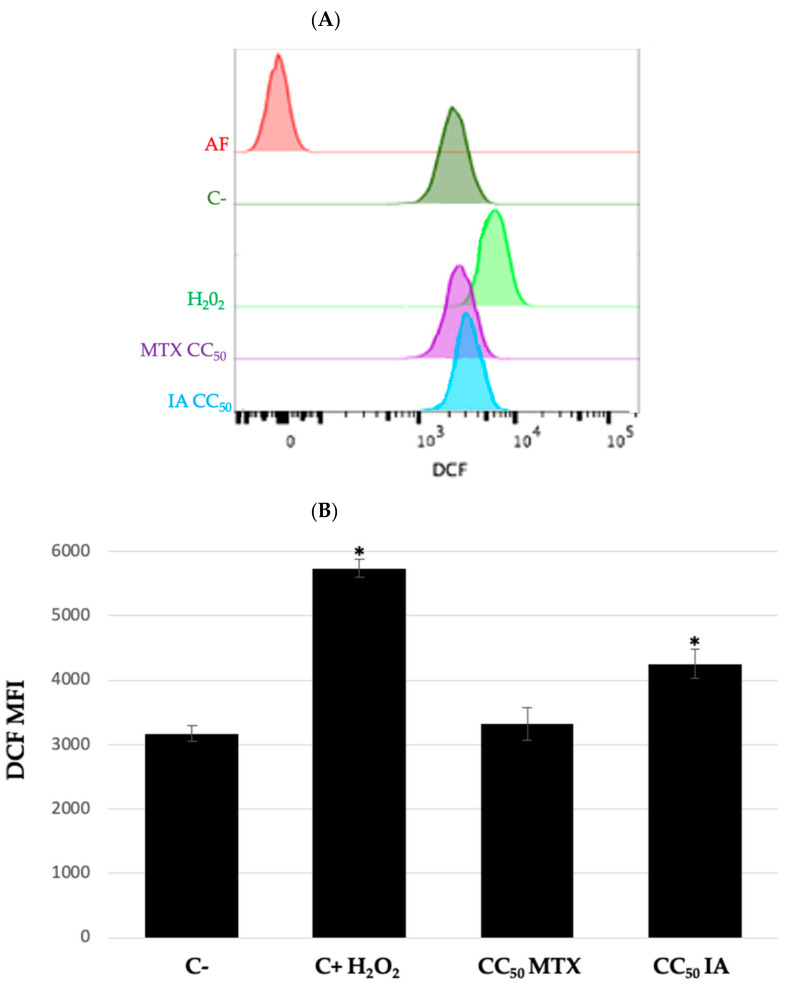
Production of ROS. Quantification was performed by luminescence using DCF reagent. Incomptine A (**IA**) increases oxygen reactive species in U-937 cells. (**A**) Representative histogram from evaluated treatments. Fluorescence peak of DCF moves towards the right depending on DCF oxidation by increasing intracellular reactive oxygen species, positive control H_2_O_2_ showed the maximum shift of fluorescence peak. (**B**) DCF index was calculated, respectively, to negative control, finding significant differences in positive control H_2_O_2_ and incomptine A treatments. Data is expressed as means ± SEM, *n* = 3; * *p* < 0.05. MTX, methotrexate; IA, incomptine A; DCF, dichlorofluorescein 2′,7′-diacetate.

**Figure 4 ijms-22-10516-f004:**
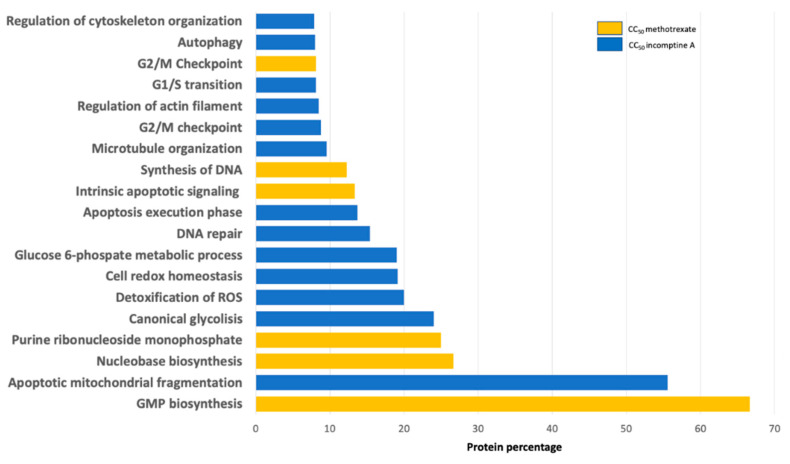
Functional enrichment analysis. Enrichment analysis revealed 82 significantly enriched biological processes and signaling pathways (*p* < 0.05). Intrinsic apoptosis can be found in incomptine A and methotrexate treatments, but different biologic processes are differentially expressed.

**Figure 5 ijms-22-10516-f005:**
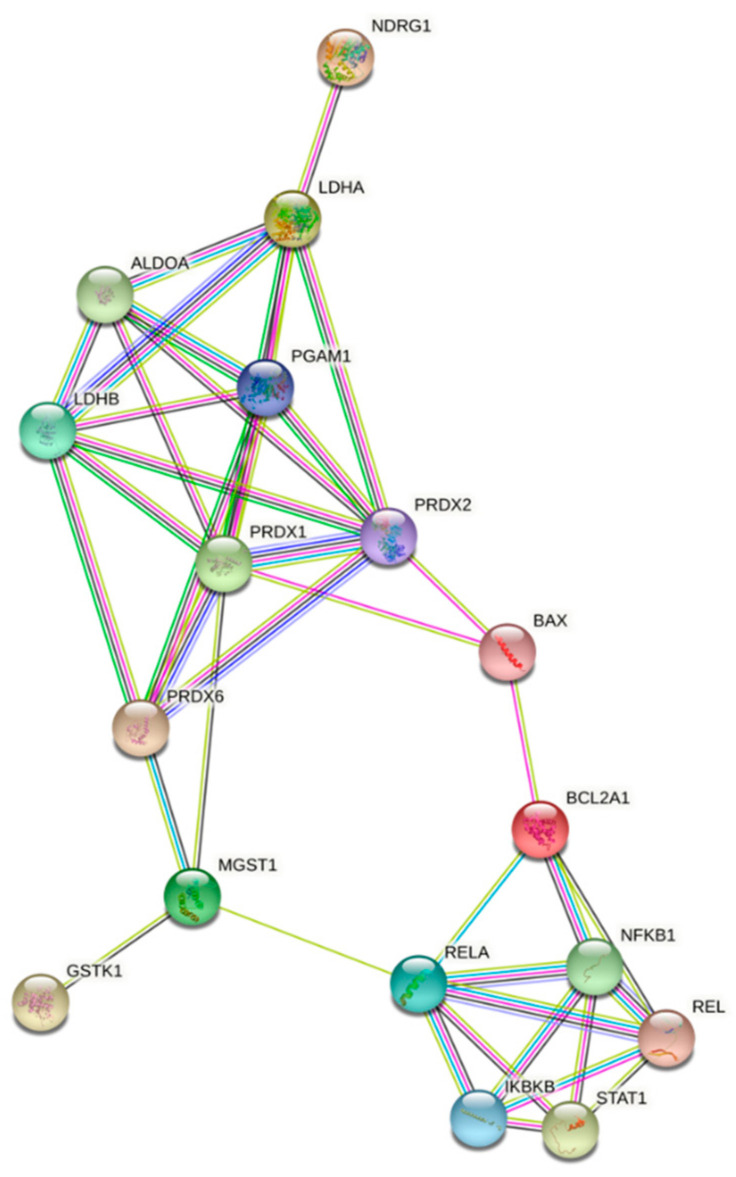
Protein–protein interaction (PPI) networks of differentially expressed proteins. Line shape indicates the predicted molecular mechanism of action. PPI presented 17 nodes and 33 edges with a PPI enrichment p-value < 1.0 meaning that proteins are biologically connected as a group.

**Table 1 ijms-22-10516-t001:** Cytotoxic activity of incomptine A in CC_50_ (mM) ± SEM against four sub-types of NHL human cell lines.

NHL Cell Line	Incomptine A (IA) CC_50_ (mM) *^a^*	Methotrexate CC_50_ (mM) *^a^*
U-937	0.12 ± 0.02	0.5 ± 0.004
Farage	2.3 ± 0.55	1.5 ± 0.01
SU-DHL-2	3.2 ± 0.04	1.3 ± 0.01
REC-1	3.5 ± 0.01	8.1 ± 0.05

*^a^* Cytotoxic concentration required to kill 50% of the cells (CC_50_). Data represent the means ± SEM, were analyzed using Graph Pad Prism, *n* = 3. NHL, non-Hodgkin’s lymphoma.

**Table 2 ijms-22-10516-t002:** Protein expression dysregulation involved in U-937 cell death induced by incomptine A (**IA**) and identified by iTRAQ. Fold change (FC) values are shown for all identified key proteins for each treatment.

Protein ID	Protein Name	Gene Name	MTX (FC)	IA (FC)
Q07812	Apoptosis regulator BAX	BAX	1.12	1.75
Q53EL6	Programmed cell death protein 4	PDCD4	1.98	2.80
Q92597	NDRG1 protein	NDRG1	1.53	1.64
Q16548	Bcl-2-related protein A1	BCL2A1	0.77	0.53
Q9NYF8	Bcl-2-associated transcription factor 1	BCLAF1	0.56	0.26
P00374	Dihydrofolate reductase	DHFR	1.42	1.77
P11586	Tetrahydrofolate synthase	MTHFD1	1.29	2.1
P60891	Ribose phosphate pyrophosphokinase 1	PRPS1	1.65	2.24
P10620	Microsomal glutathione S-transferase 1	MGST1	0.76	0.59
Q9Y2Q3	Glutathione S-transferase kappa 1	GSTK1	0.65	0.49
Q06830	Peroxiredoxin-1	PRDX1	0.93	2.74
P32119	Peroxiredoxin-2	PRDX2	1.21	1.95
P30041	Peroxiredoxin-6	PRDX6	1.27	2.29
O14920	Inhibitor of nuclear factor kappa-B kinase subunit beta	IKBKB	1.23	1.73
P19838	Nuclear factor NF-kappa-B p105 subunit	NFKB1	1.11	1.68
Q04206	Nuclear factor NF-kappa-B p65 subunit	RELA	1.79	1.62
Q04864	c-Rel	REL	1.45	1.9
P42224	Signal transducer and activator of transcription 1-alpha/beta	STAT1	1.50	1.97
P30281	G1/S-specific cyclin-D3	CCND3	1.21	0.57
P43246	DNA mismatch repair protein Msh2	MSH2	1.39	1.95
O95352	Autophagy protein 7	ATG7	1.40	2.73
Q9NT62	Autophagy protein 3	ATG3	1.31	2.21
P11279	Lysosomal associated membrane protein 1	LAMP1	0.91	1.60
P13473	Lysosomal associated membrane protein 2	LAMP2	1.47	1.86
P00338	L-lactate dehydrogenase A chain	LDHA	0.97	0.54
P07195	L-lactate dehydrogenase B chain	LDHB	0.78	0.51
P04075	Fructose-bisphosphate aldolase A	ALDOA	1.06	0.34
P18669	Phosphoglycerate mutase 1	PGAM1	1.25	0.62
O43707	Alpha actinin 4	ACTN4	1.27	0.51
Q6IRU2	Tropomyosin alpha-4 chain	TPM4	0.96	0.43

## Data Availability

The additional data on this study is available on request from corresponding author.

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
