# Peer review of "Incomptine A Induces Apoptosis, ROS Production and a Differential Protein Expression on Non-Hodgkin’s Lymphoma Cells"

_ijms, 2021, doi:10.3390/ijms221910516_

Round 1

Reviewer 1 Report

Figure 2B. I suggest to split the bar with "apoptotic cells" on two bars: early and late apoptotic cells. It will be compatible with 2A.

Figure 4. I suggest to prepare double bar graphs presenting how both compounds modify described processes. It is unlikely that tested compounds modulate them selectively. 

Figure 5. the quality of figure is low. It should be corrected.

Author Response

Mr. Gavyn Lei Assistant editor ((E-Mail: gavyn.lei@mdpi.com)/Hidayat Hussain, PhD -Gues EditorDepartment of Bioorganic Chemistry (Hidayat.Hussain@ipb-halle.de) 

ANSWER TO EDITOR AND REVIEWER

In agree with editor and reviewer we decided revise the manuscript “Incomptine A Induces apoptosis, ROS production and a Differential Protein Expression on Non-Hodgkin´s Lymphoma Cells (ijms-139406)” manuscript and made all comments from editor and reviewer these changes are in yellow color, including:

Comments and suggestion for Authors

Reviewer 1

Query 1.- Figure 2B. I suggest to split the bar with “apoptotic cells” on two bars: early and late apoptotic cells. It will be compatible with 2A

Answer

In agree with the suggestion Figure 2B was adapted.

Query 2.- Figure 4. I suggest to prepare double bar graphs presenting how both compounds modify described process. It is unlikely that tested compounds modulate them selectively.

Answer

In this case the Figure 4 was no change because, the intention of Figure 4 is to present the different biological processes that were identified with significant changes in protein expression for both treatments. It should be noted that the article does not mention that incomptine A and methotrexate treatments selectively modulates identified processes. Figure 4 presents only those processes in which alteration of the expression of at least 5% of the proteins involved in each described process was identified (for each treatment), so the suggested modification was not carried out.

Query 3.- Figure 5. The quality of figure is low. It should be corrected.

Answer.

Figure 5 quality weas improved, now it has the highest image resolution available from STRING Protein-Protein Interaction networks.  https://string-db.org/

Additional mistakes were corrected in manuscript yellow color see lines: 4, and 24,

Dr. Fernando Calzada

Reviewer 2 Report

General comment:

This article studied the cytotoxic and anticancer properties of incomptine A (IA) extracted from Decachaeta incompta. It revealed that IA exhibited cytotoxic activity against the cell line U-937 and significantly increased apoptotic cell death as well as intracellular ROS levels. Differential protein expression analysis showed that most of the differentially regulated proteins are involved in apoptosis, oxidative stress, glycolytic metabolism, or cytoskeleton structuration. Authors also claimed that the inhibition of glycolytic enzymes (ALDOA, LDHA, and LDHB) and cytoskeleton proteins (alpha actinin-4 and tropomyosin alpha-4 chain) may be part of the mechanism by which IA produce anti-cancer effects, which is interesting and potentially could contribute to the research field and therefore deserves to be published. However, the reviewer thought that the manuscript needed minor revisions before publication.

Specific comments:

Title

The title is informative and relevant to the major findings.

Abstract

In the abstract, the aim of the study clearly mentioned. However, recommendations are not adequately presented. In Line 29, what does it mean by 1?

Introduction

The research question/gap is not clearly outlined. The introduction may improve further by providing sufficient background.

Materials and Method

Well explained.

 Results and Discussion

The results are well structured and well explained. However, some abbreviation (IA) is not properly/consistently used (i.e., line 255, 262, 270)

Conclusion

The conclusion does not properly answer the aims of the study. Authors should briefly discuss major findings and key messages. The major limitation of the study is not presented. 

Author Response

Mr. Gavyn Lei Assistant editor ((E-Mail: gavyn.lei@mdpi.com)/Hidayat Hussain, PhD -Gues EditorDepartment of Bioorganic Chemistry (Hidayat.Hussain@ipb-halle.de) 

ANSWER TO EDITOR AND REVIEWER

In agree with editor and reviewer we decided revise the manuscript “Incomptine A Induces apoptosis, ROS production and a Differential Protein Expression on Non-Hodgkin´s Lymphoma Cells (ijms-139406)” manuscript and made all comments from editor and reviewer these changes are in yellow color, including:

Comments and suggestion for Authors

Reviewer 2.

Query 1.- In the abstract, the aim of the study clearly mentioned. However, recommendations are not adequately presented. In Line 29, what does it mean by 1.

Answer.

Line 29 was a mistake, should be “IA” and this has been corrected. In addition, the manuscript was sent to edition and checked by a native English-speaking.

Query 2.- Introduction: The research question/gap is not clearly outlined. The introduction may improve further by providing sufficient background.

Answer.

The introduction was improved in its redaction to provide a clear background on sesquiterpene lactones (SL) reported cytotoxic activity and highlighted that 3-methylenedihydrofuran-2(3H)-one moiety has been previously associated to cytotoxic properties of sesquiterpene lactones and it is responsible for their broad range of biological properties. Since incomptine A (IA) has this moiety, we examined its cytotoxic effects in non-Hodgkin lymphoma cell lines as well as apoptosis induction, reactive oxygen species and differential protein expression on U-937 cells. See line 59 to 71 in yellow color.

Query 3. Results and Discussion: The results are well structured and well explained. However, some abbreviation (IA) is not properly/consistently used (i.e., line 255, 262, 270)

Answer.

Abbreviations were revised and corrected. In addition, the manuscript was sent to edition and checked by a native English-speaking.

Query 4. Conclusion: The conclusion does not properly answer the aims of the study. Authors should briefly discuss major findings and key messages. The major limitation of the study is not presented.

Answer.

Conclusion was revised and corrected. See line 289 to 297.

Additional mistakes were corrected in manuscript yellow color see lines: 4, and 24,

Dr. Fernando Calzada